# A Long-Term Study on the Content of Polycyclic Aromatic Hydrocarbons in Rubber from End-of-Life Tires of Passenger Cars and Trucks

**DOI:** 10.3390/ma15197017

**Published:** 2022-10-10

**Authors:** Stefan Hoyer, Lothar Kroll, Kirsten Lippert, Albrecht Seidel

**Affiliations:** 1Department of Lightweight Structures and Polymer Technology, Chemnitz University of Technology, Reichenhainer Straße 31/33, 09126 Chemnitz, Germany; 2Biochemical Institute for Environmental Carcinogens Prof. Dr. Gernot Grimmer-Foundation, 22927 Grosshansdorf, Germany

**Keywords:** end-of-life tires, truck and passenger car tires, recycling, polycyclic aromatic hydrocarbon (PAH), REACH

## Abstract

At the European level, limits have been set (REACH) for the content of polycyclic aromatic hydrocarbons (PAH) in products with rubber and plastic components that come into contact with human skin or the oral cavity. These limit values reported in Commission Regulation (EU) 1272/2013 are of particular importance for the utilization of end-of-life tires (ELT) as recycled rubber materials for consumer applications, but a suitable analytical method has not yet been specified. On the other hand, comprehensive measurement series of the PAH content of ELT materials are scarce in the context of compliance testing against this regulation and general published PAH levels in ELT materials are often based on very different analytical methods. In the present work, the PAH content of three different rubber granulates from ELT (obtained from whole truck and passenger car tires and truck tire treads) were investigated over a period of two years. The Grimmer method was used for PAH profile analysis, which in terms of extraction intensity and sample preparation not only meets the requirements for a reliable determination of the EU priority PAH, but in addition covers a more comprehensive PAH profile. A total of 26 different PAH compounds, including the 8 EU priority PAH (REACH) and the 16 U.S. EPA priority PAH, were analyzed and their variations over time were examined to obtain reliable current data for PAH content in rubber granulates produced from ELT.

## 1. Introduction

Tires are made from rubber compounds consisting of a vast number of components, including natural and synthetic rubber, plasticizer oils, amorphous silica, and carbon black as filler and pigment as well as a variety of other additives and auxiliary materials of lesser quantity [1]. Among this complex mixture of components both plasticizer oils and carbon black are a source of polycyclic aromatic hydrocarbons (PAH) [2], a group of toxic compounds known to pose a risk to human health [3]. End-of-life tires (ELT) can be recycled into rubber granulates and powder, which could subsequently be processed into secondary products, such as elastic damping mats or other molded parts. However, in recent years, limit values for the content of PAH in such products have been adopted at the European level (REACH) based on the precautionary principle. With regard to these limits, a distinction must be made between newly produced tires on the one hand and secondary products manufactured from recycled ELT on the other.

In order to reduce the content of PAH in tires, PAH limits for extender oils used in the manufacturing of tires have been set into force since 2010 by Regulation (EC) No. 1907/2006 (REACH) [4]. The plasticizer oils currently used should not exceed a benzo[*a*]pyrene (B[*a*]P) content of 1 ppm and a sum value of 10 ppm for the eight priority EU REACH PAH (B[*a*]P, chrysene, benzo[*a*]anthracene, benzo[*b*]-, benzo[*k*]- and benzo[*j*]fluoranthene as well as benzo[*e*]pyrene and dibenzo[*a,h*]anthracene) [5]. The determination of these PAH in plasticizer oils is carried out according to the European standard DIN EN 16143. Interestingly, compliance with this requirement for the vulcanized rubber produced is verified by an indirect method, investigating an acetone extract by a high field ^1^H-NMR measurement according to the ISO 21461 method. Results of such measurement of tires according to this regulation can be found in several reports [6,7,8,9,10]. However, in addition to extender oils, the filler and reinforcing material carbon black is also an important source of PAH in tires, the content of which is not regulated at EU level. Thus, there is formally no applicable limit for the total content of PAH in tires.

The Regulation (EU) No 1272/2013 of the European Union [11], in contrast, refers to accessible parts of plastic or rubber articles that come into regular or prolonged contact with human skin or oral cavity. This field of validity also includes a series of molded parts that are made, in whole or in part, from recycled ELT material. This regulation limits the total content of the eight EU priority PAH [12] to 1 mg/kg each, but for toys and other baby and toddler products, a more strictly limit value of 0.5 mg/kg each has been set. More recently, the concentration of PAH in rubber granulates used as infill material for synthetic turf pitches or playgrounds was restricted to a maximum of 20 mg/kg for the sum of eight EU PAH (Commission Regulation (EU) 2021/1199 [13]). Since these regulations refer to the PAH content of the entire rubber material, whereas the regulation for newly manufactured tires only concerns the PAH concentration of applied extender oils, the analytical methods used and the values obtained are not comparable. 

Nevertheless, accurate knowledge of the PAH content and its variation is a key prerequisite for setting maximum recyclate contents in products falling within the scope of the various legal regulations or for adapting these regulations according to the ALARA principle (as low as reasonably achievable) [14]. Diekmann et al. [15] recently reviewed the available data for PAH in consumer goods made from ELT materials and mentioned that it is difficult to estimate the PAH content of such products derived from recycled rubber material. The levels vary between different batches and in addition it is often not possible to determine the origin of the material (truck or car tires). Different studies would therefore result in different estimates of the PAH content. 

However, Regulation (EU) No 1272/2013 has not yet specified an analytical method to be used to determine the PAH content. A report was prepared by the European Chemicals Agency (ECHA) [16] at the request of the European Commission to support the review of the corresponding paragraphs (5 and 6 of entry 50 of Annex XVII of REACH). ECHA reviewed the available methods for the determination of the PAH content in products. The analytical method developed at the Joint Research Centre (JRC) in Ispra by Geis et al. [17] (subsequently referred to as JRC method) was found to be suitable for the determination of very low concentrations of the eight EU PAH in rubber and plastic components. The European Committee for Standardization (CEN) is actually going to address the remaining shortcomings within the currently performed work to develop a harmonized analytical standard [16]. 

The JRC method is based on a continuously hot extraction according to Randall with toluene over a period of 3 h [17]. Compared to other methods frequently used in this context, such as DIN ISO 18287, this is a relatively intensive extraction method that also extracts the PAH bound to carbon black. 

Comprehensive and reliable data on the PAH content of ELT in the context of Regulation (EU) No. 1272/2013 is difficult to find in the literature. This is especially true for results obtained with intensive extraction procedures, such as hot extraction according to Randall used in the JRC method [17]. As reported by Diekmann et al. [15] in their review, the majority of measurements are based on extraction of PAH in an ultrasonic bath, one study determined the PAH content using the Grimmer method. The Grimmer method, which is also used in this work, is quite similar to the JRC method with regard to the intensity of the extraction, tending to be even more comprehensive due to the substantially longer extraction time.

The present study aimed to investigate the PAH content of three different rubber granulates from ELT (obtained from whole truck and passenger car tires as well as truck tire treads), which is relevant in the context of Commission Regulation (EU) No. 1272/2013, over a period of two years. The well established Grimmer method, selected for this study utilizes a Twisselmann extraction, which is also a continuous hot extraction like the Randall technique. A total of 26 different PAH compounds were analyzed including the 8 EU priority PAH (REACH) [12], the 16 U.S. EPA priority PAH [18], and the 22 environmentally relevant PAH in Germany (established as monitoring list by the German Federal Environmental Agency [19]) and the variations of their values was followed over two years. With the values obtained in this way, it should therefore be possible to derive an estimate of the measured levels of PAH content in ELT material, if a harmonized analytical method, possibly based on the JRC method, is established at the European level. 

## 2. Materials and Methods

### 2.1. Characterization of the Sample Material

Three different types of ELT recyclates were investigated in this work (see Table 1). These are recyclates from (a) whole truck tires (TW), (b) truck tire treads (TT) and (c) whole passenger car tires (PW), all of which were largely free of steel and textile components. “Whole” thereby means that the material was produced by granulation of the entire tire. The recyclates from truck tires were provided by Mülsener Rohstoff- und Handelsgesellschaft mbH MRH (Mülsen, Germany), and the passenger car material by PVP Triptis GmbH (Triptis, Germany). Since these companies only process truck tires (MRH), respectively, passenger car tires (PVP) with their equipment, the material can be described as single-origin. The recyclates went through several successive shredding and intermediate storage steps. It is therefore assumed that the material is sufficiently mixed and thus cannot be attributed to a single tire or even a single component of a tire. Therefore, the material is considered homogeneous and representative of the population of recyclates. However, the analyzed recyclates differed in their particle size distribution, as shown by the granulometric measurements with a laser particle sizer (Analysette 22, FRITSCH GmbH, Germany) in Figure 1. The granulates TW and PW show a similarly wide particle size distribution, with TW approximately in the range of 0.8 to 3 mm (median 1.75 mm) and PW in the range of 0.4 to 2.5 mm (median 1.25 mm). The truck tire tread material TT was only available in the form of cryogenically ground buffings from retreading, which is why there is a strongly deviating grain size of about 0 to 0.7 mm (median 0.25 mm) here. In Table 1, a distinction is made between the real grain size distribution, measured by a laser particle sizer, and the mesh size at which this material was sieved. Typically, a factor of 1.7 is found between the mesh size of the sieve and the real upper particle size of the sieved material (Figure 1).

### 2.2. Sampling Procedure

#### 2.2.1. Truck Tires, Whole (TW)

For the monthly samples, Mülsener Rohstoff- und Handelsgesellschaft mbH (MRH) took samples over a period of two weeks (production volume of approximately 100 metric tons). On several days, two to three times each 50–100 g of the current production is taken. Overall, a 2–3 kg monthly sample is created. From the monthly sample thus obtained, a sample of about 300 g is separated by means of sample splitter.

#### 2.2.2. Truck Tires, Tread (TT)

The material is produced by cryogenic grinding of buffings from truck tire retreading. Cryogenic grinding of this material is carried out at MRH Mülsen only when required and in batch sizes of approx. 22 tons with a pin mill contraplex CW 630 from Hosokawa Alpine, Germany. This batch processing minimizes contamination with material from the whole tire recycling that would otherwise be carried out, but it cannot be quantified precisely. Sampling takes place analogously to that of the whole truck tires (TW).

#### 2.2.3. Passenger Car Tires, Whole (PW)

PVP Triptis takes a sample from the current production every 3 h, just before bagging the material. The material collected on a daily basis is then reduced by means of a riffler divider into daily samples of about 150–300 g. The daily samples are then combined into weekly, and these are finally combined into monthly samples, whereby the sample is mixed intensively and its amount is reduced at each union by means of a riffler divider.

#### 2.2.4. Measurement Frequency

The measurements were carried out in the period from July 2017 to July 2019 according to Table 2. The number of repeated determinations of a respective material sample is entered in the corresponding fields. The samples have been collected over a certain period and analyzed in four discrete measurement series (each marked yellow and grey).

For passenger car tires, individual measurements of weekly samples were initially taken in the months July to September 2017 (see Table 2, week number). For truck (August 2017–July 2018) and car tires (October 2017–July 2018), the monthly mixed samples were determined twice in the first year. In the following year, only one single measurement was carried out at a time. For truck treads, one individual measurement was carried out in the first year, using cryogenically ground powder smaller than 400 µm. In June 2009 an additional triple measurement of a single sample was carried out, whereby untreated buffings from the truck tire retreading was selected as the sample form.

### 2.3. Extraction and Quantification of PAH

#### 2.3.1. Analytical Method

The PAH profile analyses of the ELT materials were performed at the Biochemical Institute of Environmental Carcinogens (BIU) Prof. Dr. Gernot Grimmer-Foundation (Grosshansdorf, Germany) applying the so-called Grimmer method. This analytical method is based on the stable isotope dilution principle using GC-MS with selected ion monitoring (SIM mode) and allows the quantification of the PAH content in the sub-ppb range. The Grimmer method has been validated for various matrices during the work of BIU for the Environmental Specimen Bank of the German Federal Environment Agency (UBA) and is published [19]. 

#### 2.3.2. Chemicals and Materials

Reference materials of naphthalene, acenaphthylene, acenaphthene, fluorene, phenanthrene, anthracene, fluoranthene, pyrene, benzo[*b*]naphtho[2,1-*d*]thiophene, benzo[*ghi*]fluoranthene, benzo[*c*]phenanthrene, benzo[*a*]anthracene, cyclopenta[*cd*]pyrene, triphenylene, chrysene, benzo[*b*]fluoranthene, benzo[*k*]fluoranthene, benzo[*j*]fluoranthene, benzo[*e*]pyrene, benzo[*a*]pyrene, perylene, indeno[1,2,3-*cd*]pyrene, dibenzo[*a,h*]anthracene, benzo[*ghi*]perylene, anthanthrene, and coronene were obtained from LGC standards.

As internal standards the following PAH compounds were used: D8-naphthalene, D8-acenaphthylene, D10-acenaphthene, 2-fluorofluorene, D10-phenanthrene, D10-fluoranthene, D10-pyrene, D12-benzo[*a*]anthracene, D12-benzo[*b*]fluoranthene, D12-benzo[*a*]pyrene, D12-benzo[*ghi*]perylene, and indeno[1,2,3-*cd*]fluoranthene. All deuterated PAH compounds and indeno[1,2,3-*cd*]pyrene were purchased from Dr. Ehrenstorfer (Augsburg, Germany) and 2-fluorofluorene was delivered by Sigma-Aldrich (Steinheim, Germany). The purity of all PAH compounds was proven to be better than 99% as checked by GC-FID. Silica gel (type 60, particle size 0.063 to 0.200 mm) was provided by MP Biomedicals (Heidelberg, Germany). Toluene and *N*,*N*-di–methylformamide (DMF) were from Honeywell (Seelze, Germany), whereas cyclohexane was purchased from Biesterfeld (Hamburg, Germany). Toluene and cyclohexane were distilled before use.

#### 2.3.3. Extraction of PAH from ELT Materials

An aliquot of the rubber granulate (3 g) is placed in a glass fiber thimble and subjected to a hot extraction with toluene for 8 h in a Twisselmann apparatus. The resulting extract is allowed to cool down to room temperature and diluted with toluene if necessary. Subsequently an aliquot of 2 mL is taken to which 2 mL of a stock solution is added containing all 12 internal PAH standards.

#### 2.3.4. Sample Preparation of the Extract

The crude extract is diluted with 50 mL cyclohexane followed by a liquid–liquid extraction with a mixture of *N*,*N*-dimethylformamide (DMF) and water. After dilution of the aqueous phase the PAH are back-extracted with cyclohexane. The combined extracts are concentrated by means of a rotary evaporator under reduced pressure to approximately 1 mL. This solution is subjected to a silica gel SPE cartridge which is eluted with cyclohexane. The obtained PAH fraction is subsequently concentrated and an aliquot of 1 µL is used for injection to the GC-MS instrument.

#### 2.3.5. Instrumental PAH analysis

The GC-MS analysis is performed using an Agilent 6890N instrument connected to an Agilent 5973N quadrupole mass spectrometer operated in the selected ion monitoring (SIM) mode and an Agilent 7683 Autosampler from Agilent Technologies (Santa Clara, CA, USA). Separation of the PAH profile is performed on an Agilent DB-35MS capillary (Agilent Technologies, 30 m × 0.25 mm i.d. × 0.25 µm film thickness, virtually equivalent to a (35%-phenyl)methylpolysiloxane) using helium (purity 99.999%) as carrier gas (flow rate 1 mL/min). Separation of PAH is achieved using the following conditions: splitless injection of 1.0 µL sample; the GC oven temperature was programmed from 100 °C (held 5 min) to 160 °C (20 °C min^−1^) to 250 °C (5 °C min^−1^) to 340 °C (3 °C min^−1^ and held 5 min). A 3-point calibration is performed for each PAH compound with linear curve fitting in a working range of 0.03 to 10 ng/µL. The coefficients of determination (R^2^) of the calibrations are compiled in Appendix A Appendix A. The limits of quantification (LOQ) and limits of detection (LOD) are determined using the signal-to-noise ratios (S/N; LOQ is determined by a S/N ratio of 10:1) and are reported for each individual PAH in Appendix A. Identification of PAH is conducted based on relative retention times and molecular ions compared to reference materials. Quantification is achieved via the PAH applied as internal standards (stable isotope dilution method). The system is operated by Agilent Enhanced ChemStation Software (G1701DA Version D.00.00.38).

## 3. Results

### 3.1. Notes on the Presentation of Results

The results of the measurements are summarized in Figure 3 and Figure 4. The results for the PAH content *w*
*_x,y_* of each material (*x*) and each individual PAH species (*y*) is presented in the form of a box plot (see Figure 2). The color of the boxplots corresponds to the font color of the different tire types in the upper left corner of each figure. In the second and third column, the eight EU PAH as well as the 16 U.S. EPA PAH are marked with an “*x*”. Table 2 and Table 3 present the derived statistical values. 

Additional remarks of the presentation and evaluation of results in Table 3 and Table 4:Each measured value of a discrete PAH species and the sum values in the three lower columns are to be considered separately. The sum values were calculated separately for each individual measurement and then the statistical characteristic values were derived from these discrete individual sums. In this respect, the sums of the discrete PAH will not necessarily correspond to the respective sum values in the lower three lines.The margin of error *e* corresponds to half the width of the two-sided confidence interval of the mean value $\bar{x}$ at the confidence level of 95% (α = 0.05). The maximum value represents, for each individual PAH or for the respective sums, the highest value measured in an individual measurement over the entire measurement series.On the basis of the *p*-value it was examined whether a normal distribution of the measured values [*w_x,y_**~ N*(*µ*, *σ^2^*)] could be excluded. The Anderson–Darling test used for this purpose compares the measured values with the theoretical distribution of the values in relation to the normal distribution. If the *p*-value is less than 0.05, the hypothesis that the values correspond to a normal distribution should be rejected (in this case the fields were marked red). In contrast, a *p*-value greater than 0.05 does not necessarily mean that the data is normal distributed. The Anderson–Darling test can be used for a sample size of *n* ≥ 8, which is why an evaluation for buffings of truck tread (*n* = 3) was omitted.In addition to testing the results for normal distribution [*w**_x,y_**~ N(µ,*
*σ^2^*)], the test for log-normal distribution [*w**_x,y_**~ LN(µ*, σ^2^**)] was also carried out. For this purpose, the data was transformed by logarithmizing each individual measurement result: *w***_x,y_* = ln(*w**_x,y_*). The corresponding fields were marked red if the *p*-value is less than 0.05, thus indicating that a log-normal distribution can be rejected.The respective higher result for the *p*-value of *w**_x,y_* or *w***_x,y_* was marked green if they were higher than 0.05.


### 3.2. Overview of the Measurement Results for all Three Analyzed Tire Materials

Figure 3 initially presents the total range of measured values for the PAH content of whole passenger car tires (red), whole truck tires (black) and truck treads (blue) in the form of boxplots. Table 3 and Table 4 show statistical indicators of the measurement series. In the evaluation of the measured values for the passenger car tires, no distinction was made between the weekly and monthly mixed samples, as outlined in Section 3.3.

### 3.3. Comparison of the Weekly and Monthly Mixed Samples from Passenger Car Tires

In Figure 4 the measurements of weekly samples of whole passenger car tires (PW, week number 30–38 of 2017) are compared with those of monthly samples (October 2017 to July 2019). It turns out that the measurements of the weekly samples do not yield considerably different results. With the exception of acenaphthene and fluorene, the measured extreme values also lie within the fluctuation ranges of the monthly samples. In this respect, it is assumed that a dilution effect of the samples, which can result from mixing the individual weekly samples to monthly samples, does not substantially change the measured values. Therefore, no distinction is made between weekly and monthly samples to derive the statistical values in Table 3 and Figure 3.

## 4. Discussion

Tires usually contain a variety of known toxic compounds among them PAH are well known to have adverse effects on humans, but can also be harmful to environmental organisms, if released from the vulcanized rubber matrix [2,20,21]. The present study was undertaken to provide current and reliable data on the PAH content of end-of-life tire crumb rubber which is used as recycled material in a variety of consumer products. The compliance with EU legislation regarding the allowed maximum values for PAH in ELT is determined by analytical measurements of the PAH concentration in such materials. The major sources of PAH content in rubber compounds and therefore also in ELT materials are the process oils used as plasticizer and the carbon black used as reinforcing filler and as pigment. The initial step of any applied analytical method, namely the comprehensive extraction of PAH from the vulcanized rubber, is therefore of crucial importance for a reliable PAH determination. Solvent extractions are common methods, but their efficiency varies depending on the type of solvent used and the extraction conditions (such as temperature or ultrasonic-assistance) [17,22,23,24]. At present the DIN ISO 18287:2006-05 method and the AfPS method (established by the German Commission on Product Safety as AfPS GS 2019:01 PAK) are frequently used for PAH determination in rubber and plastic samples. While the DIN ISO 18287 method involves a stepwise extraction with acetone and petroleum ether with shaking at room temperature (method A), the AfPS method involves an ultrasound-assisted extraction with toluene at 60 °C. Despite the fact that toluene is well known to be one of the most suitable solvents for PAH, a study by Hamm et al. [25] indicates that the AfPS method is unsuitable for PAH determination in carbon black and pointed out that a Soxhlet extraction would be required. Therefore, it was decided to use the Grimmer method with Twisselmann extraction in this study, entailing a continuous hot extraction with toluene, which is less time consuming compared to the Soxhlet technique.

In order to illustrate the significance of the methodological differences discussed, a comparison of the Grimmer method with the method according to DIN ISO 18287:2006-05 was carried out on the basis of identical sample material. The measurements were always based on whole passenger car tires (PW, Table 1) from the company PVP Triptis GmbH. Five samples (February to June 2019) were examined, all taken from the identical monthly sample in each case and a single measurement carried out for each. The sample material as well as the underlying sampling and the preparation of the monthly mixed samples was identical to Section 2.1. (passenger car tires, PVP Triptis GmbH). As indicated by the compiled data in Table 5, the application of an intensive sample extraction as in the Grimmer method leads to higher values for the PAH content than in the DIN ISO 18287:2006-05 method.

In general, this indicates that a more comprehensive extraction of PAH from rubber, as ensured by the Grimmer or JRC method, leads to a more reliable measurement of total PAH content in ELT materials. A comparison of results of the Grimmer method used in this study with previously published results of the PAH content in ELT materials is made difficult, in particular by the very different extraction methods used (see Diekmann et al. [15] and references therein). In principle, it appears problematic to use polar solvents such as acetone (DIN ISO 18287 method A) or ethyl acetate at low temperatures, regardless of whether additional ultrasonic-assistance is applied [26], since these solvents cause potentially incomplete desorption of the higher molecular weight PAH from the carbon black fraction in the rubber, in particular, and thus result in too low PAH contents in the extracts. Among the more suitable nonpolar solvents such as cyclohexane, benzene, or toluene, the latter appears to be the solvent of choice due to its high efficiency and to its lower toxicity than benzene [27].

The subsequent treatment of the obtained extracts is also of crucial importance. Whereas the JRC method relies on molecular-imprinted antibodies for further processing of the extracts, primarily allowing valid determination of the eight EU priority PAH, the Grimmer method applied in this study is well established to determine a broad spectrum of PAH, including the 16 U.S. EPA priority compounds and the eight EU priority PAH, as well as the German Environment Agency monitoring list for environmental matrices.

The results of the present study (Figure 3) reveal that for the sum of the eight EU PAH, whole truck tires have the lowest PAH content, while whole passenger car tires and truck tire treads show a significantly higher level, with their content values being quite similar. For the sum of the 16 U.S. EPA PAH and all 26 compounds, the PAH content of truck tire treads is significantly higher when compared with both whole tires, which are quite similar here. However, since the material of the truck tire treads (TT) was only available in the form of significantly finer milled material (see Figure 1), no direct comparison of the measured values with those of both whole tires (TW, PW) is to be made. With regard to the differences in the measurement results between the cryo powder and the buffings (Table 4), there are indications that the particle size of the tested material has an influence on the measurement results. This could be the result of an influence of the smaller particle size, and thus shorter migration paths, on the extraction or also the result of the significantly higher number of particles for an identical sample mass. However, the total number of investigated samples was too small in this project to draw significant conclusions, but this issue warrants further investigations.

The highest exceedance of the PAH limits according to EU 1272/2013 (1 mg/kg) for all three types of tires is found for benzo[*e*]pyrene (B[*e*]P), closely followed by benzo[*a*]pyrene (B[*a*]P). On average, the measured values for B[*e*]P are 3.0, 2.1, 2.8 mg/kg (PW, TW, TT), the maxima of all individual measurements are 3.7, 2.4, 3.6 mg/kg respectively. It is of interest to note that these findings are supported by a study by Depaolini [28], which reports similar B[*a*]P and B[*e*]P values as well as similar proportions of both PAH in ELT. In the case of passenger car tires in particular, some measurements exceeded the limit values for a number of the other EU PAH, although the arithmetic mean of the whole measurement campaign is usually just below the limits.

With regard to the characteristics of the observed variations in the results for the individual PAH compounds, an inhomogeneous picture emerges. Some PAH species show characteristics of a normal or log-normal distribution (see Table 3 and Table 4), but with partly significant differences between the individual tire types (PW, TW, TT). At this point, the obtained data does not allow any detailed statements to be made about the distribution of the measured values beyond ruling out the existence of a (log-)normal distribution for some of the PAH based on the Anders–Darling test.

## 5. Conclusions and Outlook

This study, for the first time reports individual PAH data for three different ELT materials obtained from well-defined tire sources, including whole truck and passenger car tires as well as truck tire treads.

Based on the results of this study it can be assumed that if an analytical method similar to the Grimmer or JRC method will be introduced as a European harmonized analytical norm, a substantially higher PAH content will be determined for ELT materials compared to the DIN ISO 18287 or AfPS method, which are frequently used today. This could also mean that currently manufactured ELT products, which typically contain 90% or more ELT material, will no longer be able to comply with existing EU limit values for PAH in relevant fields of application.

The question of whether a migration limit is more reliable as an indicator of potential risk to human health or the environment than a precautionary PAH limit in the product is the subject of ongoing scientific consideration and requires further investigation [14,29]. It is reasonable to assume that it is not so much the PAH content in a product that is relevant for assessing the health risk, but rather the quantities that are released via skin contact and absorbed by the consumer.

Nevertheless, in 2013, the EU Commission has established a restriction on the content of the eight priority PAH (REACH) in plastic and rubber components of articles used by the general public under the condition that the PAH content in products should be reduced as far as reasonably achievable (ALARA). Actually, the risks addressed under the restriction should refer to the dermal doses of PAH resulting from migration from the rubber and plastic components of articles. However, at that time, there were no reliable migration methods for determining the PAH released from these articles. In contrast, however, there were already suitable analytical methods available for the PAH content determination in such articles, which made it possible to enforce the restriction on the basis of a content limitation. In the meantime, the methods for measuring the migration of PAH from products have been further developed, so that their transfer into a liquid simulant mimics the migration into human skin as realistically as possible [30,31]. The available data from recent studies indicate that the migration rates of PAH from ELT products are very low [16,29,32].

Whether the current PAH content limit provides sufficient protection against health risks from PAH exposure, or whether a migration limit could be set for this purpose as an exception to or in addition to the PAH content limit, would require further risk assessment, as noted by the ECHA [16]. To this end, the measurement results presented here are thought to be of interest for future discussions. A scientifically based statement would be desirable in any case and urgently required in view of the importance to protect human health and reduce environmental contamination. Likewise, this would be of considerable interest for the legal and planning security of recycling companies as well as the expansion of the circular economy.

The results of the present study strongly suggest that further studies on the variation of the individual PAH levels are needed in order to determine statistically validated maximum values for the eight EU priority PAH in the different types of ELT. In this context, the detailed measurement results are attached as Appendix A (Appendix A). In some cases, a (log-)normal distribution seems to be present to a good approximation, but with respect to the critical PAH compounds B[*a*]P and B[*e*]P, this does not seem to be the case, which is why a careful analysis seems necessary to determine the highest PAH concentration to be assumed. Based on such results, maximum admixture rates of ELT material to new materials may subsequently be determined, or the existing limit values could probably be adjusted considering the ALARA principle.

## Figures and Tables

**Figure 1 materials-15-07017-f001:**
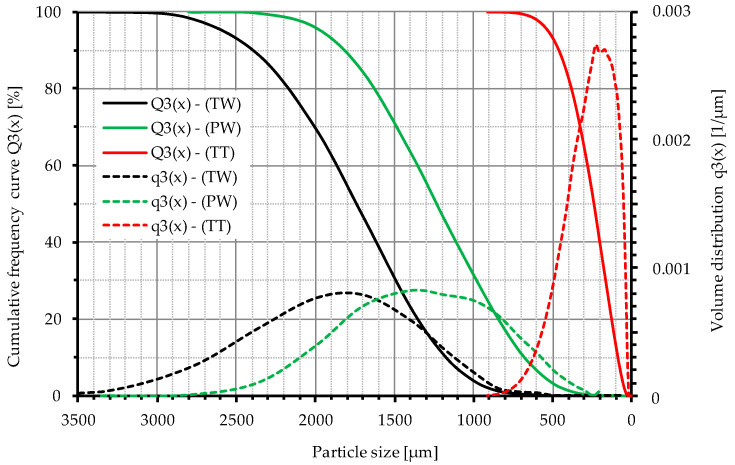
Particle size distribution of rubber recyclates from end-of-life tires, TW: whole truck tires, TT: truck tire tread, PW: passenger car tires.

**Figure 2 materials-15-07017-f002:**
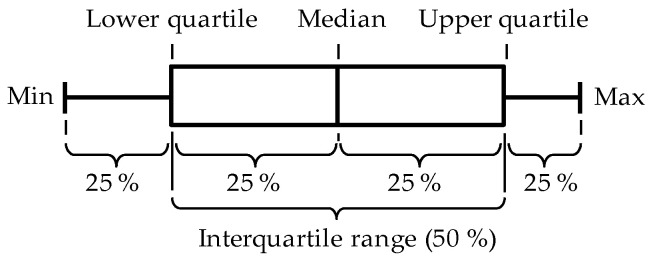
Notes on the presentation of results in form of a box plot.

**Figure 3 materials-15-07017-f003:**
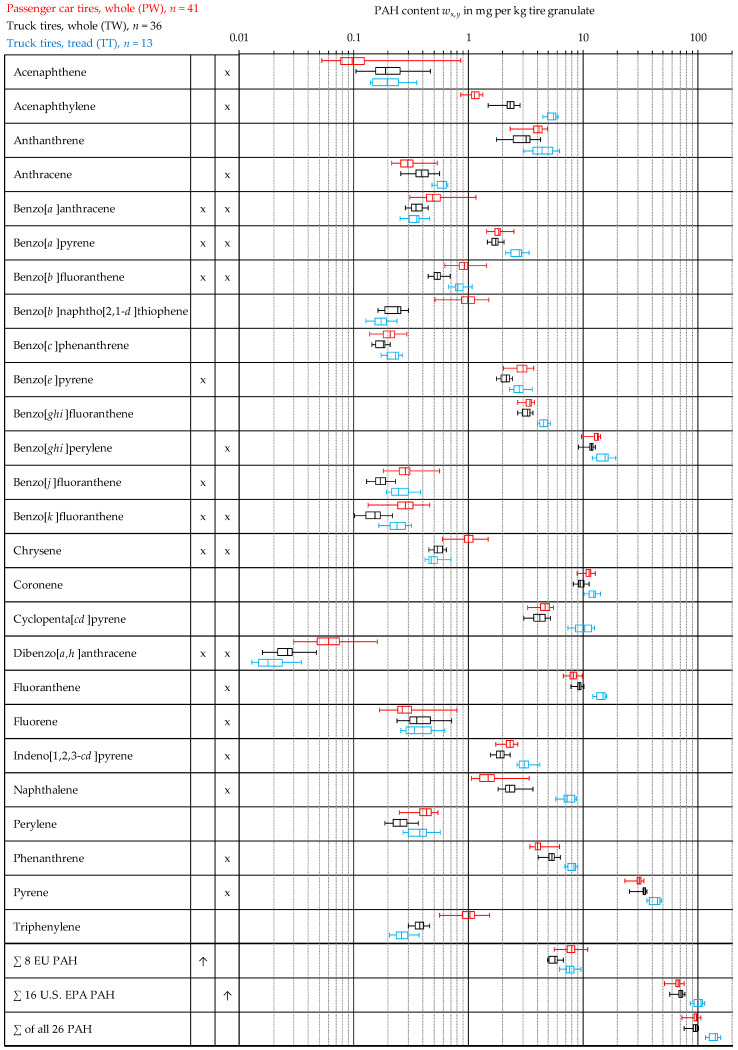
Measurement data distribution of the entire measurement series for all three materials investigated. For truck tire treads, the results of the triple measurement of buffings were not taken into account.

**Figure 4 materials-15-07017-f004:**
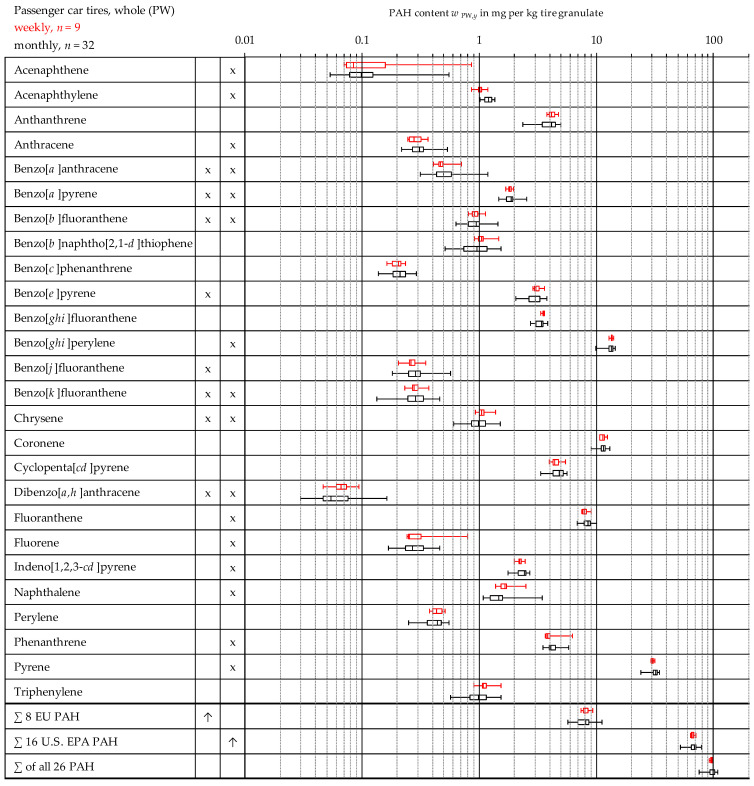
Overview of the measurement data distribution of the entire measurement series for weekly and monthly mixed samples from whole passenger car tires in the form of a boxplot (see Section 3.1.).

**Table 1 materials-15-07017-t001:** Mesh size and supplier of the three end-of-life tire materials.

	Truck Tires, Whole (TW)	Truck Tires, Tread (TT)	Passenger Car Tires, Whole (PW)
Grain size	3000–800 µm	700–0 µm	2500–400 µm
Mesh size	2000–500 µm	<400 µm	2000–500 µm
Supplier	MRH	MRH	PVP

**Table 2 materials-15-07017-t002:** Measurement frequency of the different materials.

Year	2017						2018												2019							∑
Month	7	8	9	10	11	12	1	2	3	4	5	6	7	8	9	10	11	12	1	2	3	4	5	6	7	
Truck tires, whole (TW)	/	2	2	2	2	2	2	2	2	2	2	2	2	1	1	1	1	1	1	1	1	1	1	1	1	36
Truck tires, tread (TT)	1	1	1	1	1	1	1	1	1	1	1	1	1	/	/	/	/	/	/	/	/	/	/	3*	/	13/3
Car tires, whole (PW)	2	4	3	2	2	2	2	2	2	2	2	2	2	1	1	1	1	1	1	1	1	1	1	1	1	41
	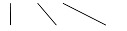																									
Week number	30–31		32–35		36–38		* Buffings																			

**Table 3 materials-15-07017-t003:** Statistical indicators for whole passenger car tires (PW) and whole truck tires (TW).

	PW: Passenger Car Tires, Whole (*n* = 41)						TW: Truck Tires, Whole (*n* = 36)					
	Arithmetic Mean $\bar{x}$	Margin of Error *e*	Sample Standard Deviation *s*	Maximum Value	*p*-Value *w*_PW, *y*_	*p*-Value for *w**_PW, *y*_ = ln(*w** *_PW, *y*_)	Arithmetic Mean $\bar{x}$	Margin of Error *e*	Sample Standard Deviation *s*	Maximum Value	*p*-Value *w*_TW, *y*_	*p*-Value for *w**_TW, *y*_ = ln(*w* _TW, *y*_)
Acenaphthene	0.13	0.04	0.14	0.87	0.00	0.00	0.21	0.03	0.08	0.46	0.02	0.51
Acenaphthylene	1.15	0.04	0.12	1.35	0.68	0.56	2.34	0.09	0.27	2.82	0.17	0.05
Anthanthrene	3.92	0.22	0.69	4.88	0.00	0.00	2.99	0.22	0.64	4.27	0.01	0.00
Anthracene	0.31	0.02	0.06	0.54	0.08	0.67	0.39	0.02	0.07	0.56	0.44	0.34
Benzo[*a*]anthracene	0.51	0.05	0.15	1.18	0.00	0.27	0.36	0.02	0.05	0.45	0.10	0.24
Benzo[*a*]pyrene	1.82	0.06	0.19	2.50	0.02	0.01	1.72	0.05	0.14	2.04	0.81	0.66
Benzo[*b*]fluoranthene	0.92	0.05	0.16	1.44	0.34	0.28	0.54	0.02	0.06	0.69	0.05	0.17
Benzo[*b*]naphtho[2,1-*d*]thiophene	0.99	0.08	0.26	1.53	0.36	0.02	0.23	0.01	0.04	0.30	0.00	0.00
Benzo[*c*]phenanthrene	0.21	0.01	0.03	0.29	0.93	0.66	0.17	0.01	0.02	0.21	0.01	0.00
Benzo[*e*]pyrene	2.96	0.14	0.43	3.74	0.32	0.06	2.10	0.07	0.20	2.43	0.07	0.04
Benzo[*ghi*]fluoranthene	3.34	0.09	0.29	3.77	0.00	0.00	3.23	0.09	0.26	3.64	0.00	0.00
Benzo[*ghi*]perylene	12.95	0.37	1.16	14.31	0.00	0.00	11.58	0.34	1.01	12.73	0.00	0.00
Benzo[*j*]fluoranthene	0.29	0.02	0.07	0.56	0.01	0.19	0.17	0.01	0.02	0.23	0.35	0.63
Benzo[*k*]fluoranthene	0.29	0.02	0.06	0.46	0.89	0.46	0.15	0.01	0.03	0.22	0.80	0.37
Chrysene	1.01	0.06	0.20	1.49	0.63	0.43	0.55	0.02	0.05	0.64	0.07	0.12
Coronene	11.22	0.28	0.87	12.82	0.17	0.05	9.60	0.25	0.74	11.25	0.41	0.61
Cyclopenta[*cd*]pyrene	4.59	0.20	0.62	5.52	0.02	0.00	4.16	0.22	0.64	5.22	0.56	0.40
Dibenzo[*a,h*]anthracene	0.07	0.01	0.03	0.16	0.00	0.31	0.03	0.00	0.01	0.05	0.09	0.59
Fluoranthene	8.26	0.22	0.70	9.94	0.54	0.70	9.18	0.21	0.63	10.17	0.00	0.00
Fluorene	0.29	0.03	0.11	0.79	0.00	0.03	0.39	0.04	0.12	0.72	0.01	0.18
Indeno[1,2,3-*cd*]pyrene	2.30	0.07	0.22	2.69	0.27	0.17	1.90	0.07	0.20	2.32	0.63	0.67
Naphthalene	1.62	0.16	0.52	3.39	0.00	0.00	2.36	0.12	0.35	3.64	0.05	0.29
Perylene	0.42	0.02	0.07	0.55	0.36	0.02	0.26	0.01	0.04	0.36	0.25	0.30
Phenanthrene	4.19	0.18	0.58	6.29	0.00	0.00	5.34	0.15	0.44	6.29	0.57	0.30
Pyrene	30.39	0.88	2.77	34.10	0.00	0.00	32.88	1.11	3.28	36.10	0.00	0.00
Triphenylene	1.02	0.07	0.23	1.54	0.56	0.24	0.37	0.01	0.04	0.46	0.77	0.43
∑ 8 EU PAH	7.87	0.37	1.16	11.10	0.35	0.09	5.62	0.17	0.51	6.71	0.08	0.06
∑ 16 U.S. EPA PAH	66.21	1.76	5.57	77.10	0.00	0.00	69.92	1.95	5.78	77.01	0.00	0.00
∑ of all 26 PAH	95.18	2.53	8.02	107.4	0.00	0.00	93.22	2.61	7.70	102.1	0.00	0.00

**Table 4 materials-15-07017-t004:** Statistical indicators for truck tire tread (TT).

	TT: Truck Tire Tread, Cryo Powder (*n* = 13)						TT: Truck Tread, Buffings (*n* = 3)			
	Arithmetic Mean $\bar{x}$	Margin of Error *e*	Sample Standard Deviation *s*	Maximum Value	*p*-Value for *w** *_TT, *y*_	*p*-Value for *w*** *_TT, *y*_ = ln(*w** *_TT, *y*_)	Arithmetic Mean $\bar{x}$	Margin of error *e*	Sample standard deviation *s*	Maximum value
Acenaphthene	0.21	0.04	0.06	0.36	0.26	0.49	0.10	0.01	0.01	0.11
Acenaphthylene	5.35	0.33	0.54	6.10	0.53	0.44	4.69	1.09	0.44	5.06
Anthanthrene	4.55	0.62	1.02	6.26	0.53	0.55	1.72	0.58	0.23	1.99
Anthracene	0.58	0.04	0.06	0.67	0.35	0.24	0.41	0.08	0.03	0.44
Benzo[*a*]anthracene	0.34	0.03	0.06	0.46	0.87	0.90	0.24	0.10	0.04	0.29
Benzo[*a*]pyrene	2.68	0.22	0.37	3.38	0.59	0.61	1.85	0.56	0.23	2.10
Benzo[*b*]fluoranthene	0.84	0.07	0.11	1.08	0.88	0.95	0.63	0.19	0.07	0.72
Benzo[*b*]naphtho[2,1-*d*]thiophene	0.18	0.02	0.03	0.24	0.75	0.90	0.24	0.09	0.04	0.28
Benzo[*c*]phenanthrene	0.22	0.02	0.03	0.27	0.28	0.18	0.19	0.01	0.00	0.20
Benzo[*e*]pyrene	2.79	0.22	0.36	3.59	0.66	0.77	2.15	0.45	0.18	2.35
Benzo[*ghi*]fluoranthene	4.56	0.24	0.40	5.21	0.30	0.32	3.45	0.11	0.04	3.49
Benzo[*ghi*]perylene	15.12	1.29	2.14	19.27	0.55	0.57	10.67	2.12	0.86	11.61
Benzo[*j*]fluoranthene	0.26	0.03	0.06	0.39	0.33	0.46	0.18	0.05	0.02	0.20
Benzo[*k*]fluoranthene	0.24	0.03	0.05	0.32	0.69	0.79	0.16	0.06	0.03	0.19
Chrysene	0.51	0.05	0.08	0.71	0.04	0.09	0.54	0.26	0.10	0.66
Coronene	12.16	0.76	1.26	14.27	0.50	0.39	8.15	1.28	0.51	8.74
Cyclopenta[*cd*]pyrene	10.24	1.07	1.77	12.60	0.15	0.13	7.05	2.40	0.96	7.94
Dibenzo[*a,h*]anthracene	0.02	0.00	0.01	0.04	0.19	0.53	0.00	0.00	0.00	0.00
Fluoranthene	14.43	0.83	1.38	15.98	0.12	0.10	11.38	1.29	0.52	11.72
Fluorene	0.40	0.08	0.13	0.63	0.09	0.23	0.25	0.02	0.01	0.26
Indeno[1,2,3-*cd*]pyrene	3.15	0.26	0.43	4.22	0.24	0.43	1.65	0.43	0.18	1.85
Naphthalene	7.48	0.57	0.95	8.78	0.47	0.50	6.74	1.11	0.45	7.16
Perylene	0.38	0.05	0.09	0.57	0.39	0.53	0.25	0.06	0.03	0.27
Phenanthrene	8.09	0.40	0.66	9.00	0.15	0.12	6.96	0.97	0.39	7.40
Pyrene	42.52	2.65	4.38	47.82	0.04	0.04	32.23	2.17	0.87	32.74
Triphenylene	0.27	0.03	0.05	0.37	0.24	0.54	0.30	0.08	0.03	0.33
∑ 8 EU PAH	7.69	0.57	0.94	9.57	0.80	0.78	5.76	1.61	0.65	6.51
∑ 16 U.S. EPA PAH	102.0	6.31	10.44	115.0	0.08	0.08	78.50	9.66	3.89	82.3
∑ of all 26 PAH	137.6	8.89	14.71	157.0	0.08	0.08	102.2	14.24	5.73	108.1

**Table 5 materials-15-07017-t005:** Comparison of PAH content for different methods and identical material (*n* = 5).

	Benzo[*a*]pyrene			∑ 16 U.S. EPA PAH		
	$\bar{x}$ ^1^	*s* ^2^	Min/Max ^3^	$\bar{x}$ ^1^	s ^2^	Min/Max ^3^
Grimmer method ^4^	1.61	0.19	1.45/1.9	55.06	3.39	51.1/60.2
DIN ISO 18287 ^5^	0.72	0.08	0.6/0.8	39.16	5.17	36.2/47.1

^1^ arithmetic mean, ^2^ sample standard deviation, ^3^ maximum and minimum of five measurements, ^4^ this study, ^5^ DIN ISO 18287:2006-05 carried out at SGS FRESENIUS GmbH Dresden.

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
