# Peer review of "A Long-Term Study on the Content of Polycyclic Aromatic Hydrocarbons in Rubber from End-of-Life Tires of Passenger Cars and Trucks"

_materials, 2022, doi:10.3390/ma15197017_

Round 1
Reviewer 1 Report
This paper, from Stefan Hoyer et al., investigates the presence of polycyclic aromatic hydrocarbons in rubber and plastic components over a long period (2 years) in order to find a suitable method for their analysis.
In my opinion, this work studies an interesting topic because, nowadays, the polymer waste management is the hottest topic in polymer science.
The proposed method for the extraction and analysis of PAH gives reasonable results, and it could be used as standard in future after that more data will be collected and compared. In my opinion, the work could be published after a minor revision. I would like the authors to consider the following comments before recommending publication in Materials journal:
1. The particle size distribution in Figure 1 is only mentioned at line 130, please comment the differences among the three curves. Is there a reason in the process for obtaining the particle?
2. The authors correctly wrote that the particle size has an effect on the measurements. Is there also an effect on the extraction process?
3. The readers may be interested in knowing the cost of the extraction process and if it is possible to recycle the solvent used, could you comment on this please?
4. Could it be possible to compare the results with other methods for determination of PAH (e.g., Celeiro, Maria, et al. "Investigation of PAH and other hazardous contaminant occurrence in recycled tyre rubber surfaces. Case-study: restaurant playground in an indoor shopping centre." International Journal of Environmental Analytical Chemistry 94.12 (2014): 1264-1271)?
Minor-corrections and typos:
- I would remove “, on the other hand” at line 57 because there is no mention to the first hand before in the text.
- Line 129 “Fritsch Analysette 22”, I usually would add the country of origin, e.g., Germany.
Author Response
1. The particle size distribution in Figure 1 is only mentioned at line 130, please comment the differences among the three curves. Is there a reason in the process for obtaining the particle?
Corresponding explanations were added starting from line 130. Table 1 was extended with the exact values of the grain size analysis (the values given previously only corresponded to the spezification of the mesh size from the material suppliers).
2. The authors correctly wrote that the particle size has an effect on the measurements. Is there also an effect on the extraction process?
Corresponding statements have been added, starting in line 378.
3. The readers may be interested in knowing the cost of the extraction process and if it is possible to recycle the solvent used, could you comment on this please?
It is the opinion of all authors of this manuscript that a scientific publication should not discuss the costs of analytical methods or parts thereof. The question of solvent recycling is not of particular importance for the method used in this study, since consumption is not particularly high due to the cyclic extraction process and, as with many other methods, if solvents are not used as a mixture, they can be recovered to a large extent. However, this is in general a modern laboratory standard and does not require any special comment.
4. Could it be possible to compare the results with other methods for determination of PAH (e.g., Celeiro, Maria, et al. "Investigation of PAH and other hazardous contaminant occurrence in recycled tyre rubber surfaces. Case-study: restaurant playground in an indoor shopping centre." International Journal of Environmental Analytical Chemistry 94.12 (2014): 1264-1271)?
A comparison of results of the Grimmer method used in this study with previously published results of the PAH content in ELT materials is made difficult in particular by the very different extraction methods used. This issue has now been discussed in more detail and included in the chapter Discussion considering also the given reference by this reviewer. Corresponding comments and reference to the recommended source were added to the manuscript, beginning at line 351.
5. Minor-corrections and typos:
a) I would remove “, on the other hand” at line 57 because there is no mention to the first hand before in the text.
Line 57 changed to "on contrast".
b) Line 129 “Fritsch Analysette 22”, I usually would add the country of origin, e.g., Germany.
Line 130 was changed according to the suggestion.
Reviewer 2 Report
The present study deals with the Polycyclic Aromatic Hydrocarbons (PAHs) content of three different end of life tires (ELT). Over two years samples collected and analyzed. The results of PAHs values could estimate the PAHs content in ELT.
In general, the writing is clear and without grammatical errors. . However, a few details of the experimental process need to be clarified for better presentation of the study.
- The sampling procedure, was carried out according to a standard method? If, yes please clarify and describe the process.
- The process of the cryogenic grinding of TT is according to a standard method? If yes, please clarify and describe the procedure.
- In section Instrumental PAH analysis, there is no details about the instrument conditions (temperatures, ramp program, split/spitless etc). Please report the conditions.
- At same section, please report the points of 3 point calibration in a table for each PAH.
- Please clarify the LOD and LOQ of PAHs measurement method.
Author Response
1. The sampling procedure, was carried out according to a standard method? If, yes please clarify and describe the process.
Sampling was not subject to any specific standard. It was carried out within the framework of the company-specific practice of sampling for the preparation of reserve samples. Sampling and preparation of monthly samples was carried out by the companies themselves.
2. The process of the cryogenic grinding of TT is according to a standard method? If yes, please clarify and describe the procedure.
Cryogenic grinding does not follow any standard. It is a large-scale industrial process in the context of the disposal of used tires at the company MRH Mülsen. Information on the equipment used has been added in line 154.
3. In section Instrumental PAH analysis, there is no details about the instrument conditions (temperatures, ramp program, split/spitless etc). Please report the conditions.
The manuscript has been edited to include the appropriate information (Chapter 2.3.5., beginning with line 229).
4. At same section, please report the points of 3 point calibration in a table for each PAH
The manuscript was supplemented with the corresponding data (Chapter 2.3.5., from line 229) and supplementary data in the form of calibration parameters (coefficient of determination R2) was added to the supplementary materials (Table S1).
5. Please clarify the LOD and LOQ of PAHs measurement method
The manuscript was supplemented with the corresponding data (Chapter 2.3.5., from line 229) and supplementary data was added to the supplementary materials (Table S1).